# Sensory Evaluation of Plant-Based Meat: Bridging the Gap with Animal Meat, Challenges and Future Prospects

**DOI:** 10.3390/foods13010108

**Published:** 2023-12-28

**Authors:** Swati Kumari, Amm Nurul Alam, Md. Jakir Hossain, Eun-Yeong Lee, Young-Hwa Hwang, Seon-Tea Joo

**Affiliations:** 1Division of Applied Life Science (BK21 Four), Gyeongsang National University, Jinju 52852, Republic of Korea; swatikumari0724@gmail.com (S.K.); alam6059@yahoo.com (A.N.A.); jakir1173@gmail.com (M.J.H.); ley9604@gmail.com (E.-Y.L.); 2Institute of Agriculture & Life Science, Gyeongsang National University, Jinju 52852, Republic of Korea; philoria@hanmail.com

**Keywords:** sensory evaluation methods, plant-based protein alternatives, consumer acceptance, challenges, animal meat

## Abstract

Globally, the demand for plant-based meat is increasing rapidly as these products are becoming quite popular among vegans and vegetarians. However, its development is still in the early stage and faces various technological challenges; the imitation of the sensory profile of meat is the most challenging part as these products are meant to be an alternative to animal meat. The development of a product similar to meat requires accurate selection of ingredients and processing techniques. An understanding of the relevant sensory profile can help in constructing products and technologies that are consumer-centric and sustainable. In this review, we focus on the comparative differences in the sensory profiles of animal meat and plant-based meat alternatives, particularly regarding the color, texture, and flavor, along with the methods used to compare them. This paper also explains the sensory evaluation and how it affects consumer preference and acceptability. Additionally, a direction for further research on developing better plant-based meat products is suggested.

## 1. Introduction

Alongside population increase and a shift in eating habits, plant-based meat alternatives (PBMAs) are gaining enormous attention from consumers around the world. According to a report by Bloomberg in 2021 [1], the plant-based food market is expected to reach a value of up to USD 162 billion by 2030 from USD 29.4 billion in 2020 globally as the concept of flexitarian (a casual vegetarian) is becoming quite popular among the younger generations with 14% identifying as vegetarian or vegan and 15% as flexitarian, which represents approximately 29% of the total population [2]. Additionally, people who are suffering from different health-related issues like lactose intolerance or malabsorption, or high cholesterol intake, are also finding their way to plant-based alternative products [3]. 

Plant-based alternatives are products that are created to replicate the sensory and quality characteristics of animal-based food products [4,5]. Currently, there is a great variety of alternative products available on the market with the biggest percentage represented by milk alternatives (15%) followed by plant-based meat (1.4%) in the USA [6]. This means that there is considerable room for technological and formulation development with the aim of creating a high-quality alternative to meat. The general composition of a meat alternative is 50–80% water content, 10–25% textured protein, 4–20% non-textured protein, 3–10% flavor additives, 0–15% fat content, 1–15% binding agents, and 0–5% coloring agents. All these nutritional and non-nutritional components provide textural and sensorial characteristics to the end product. However, recipe formulation and technological advancement are becoming more and more competitive in the attempt to produce a better product. Many food companies and researchers are trying to find more suitable ways to produce alternatives, not only for meat but also for fish, milk, cheese, and yogurt, with additional health-promoting properties. However, it is exceedingly difficult to replicate the texture and flavor of meat due to differences in the molecular and physiochemical properties of plants and animals as well as the lower protein efficiency and lack of nutrients in plant materials.

Apart from production, the acceptance of a novel plant-based alternative by the consumer is also necessary to replace meat in everyday dietary intake. The overall acceptance of a product does not depend solely upon its sensory properties but is also influenced by ethical aspects, personal beliefs, and the general awareness of consumers. However, sensory properties play an important role in the acceptance of any product before and during consumption. Before consumption, appearance, color, and shape have greater influence on the purchasing behavior of consumers while afterwards, the taste and texture profile matter most. According to a survey in 2021, about 82% of consumers believe that the most influential factor when purchasing a product is taste, followed by cost (66%), health (58%), convenience (52%), and lastly, environmental sustainability with 31%. In reference to these data, an unpalatable or surprising taste is the main barrier to adoption of any alternative product [7]. To predict the acceptance and preference patterns of consumers for any product, sensory evaluation is necessary. Sensory evaluation allows researchers to analyze the preferences of a consumer regarding a product which aids further formulation through incorporating different seasoning ingredients, spices, and other additives. The selection of ingredients and processing method is also crucial for optimizing the desirable sensory properties of the final product [8].

This review aims to provide a better understanding of the sensory differences between plant-based meat alternatives and animal meat by comparing their color, texture, and flavor profiles. It also discusses the techniques used in their development and the sensory evaluation used to identify the consumer-preferred alternative in both a qualitative and quantitative way.

## 2. Color Characteristics of Animal and Plant-Based Meat

Color has always been an important trait of meat products as it reflects the freshness and quality of meat. The color of meat depends on two attributes, i.e., the achromatic (without color) and the chromatic attributes which can be measured in terms of the reflectance (absorption and scattering) on the surface of the meat [9]. This absorption of light by the meat’s surface depends upon the combination of hemoglobin, cytochrome c oxidase, and mainly, myoglobin. Myoglobin and its oxidative states oxy, deoxy, and metmyoglobin demonstrate different coloring such as bright red, darker purple, and discoloration or brown, respectively. Myoglobin protein containing heme is responsible for the bright red color, while during cooking myoglobin undergoes denaturation (75 °C) and produce a brown-colored end product [10,11,12,13]. However, the achromatic factor affects the color of meat due to differences in the scattering of light caused by variance in the physical and structural properties of muscle. The scattering of light is determined by three major mechanisms on the surface of meats: (1) differences in the composition of protein and distribution of sarcoplasm (bound or free floating), (2) the light of the sarcomere, (3) spacing between the myofilament and the myofibrils; the latter two mechanisms cause a change in the diameter of myofibrils [13]. Meanwhile, plant-based alternatives are often unappealing grey products and do not have a bright red color appearance, as shown in Figure 1 [14]. Depending upon the manufacturer, it can only acquire limited shades like reddish pink or brown.

For a product to be a viable meat alternative, similar color before and after cooking should be delivered as this affects the purchasing intent of the consumer [15]. To improve the color profile of meat alternatives, different attempts have been made. For decades, synthetic food coloring agents have also been widely used in PBMAs. However, consumers find these undesirable due to side effects including toxicity, allergies, and neurocognitive effects. Hence, naturally occurring pigments like anthocyanin, carotenoids, and curcuminoids are used as coloring pigments. Initially, heat-stable coloring agents like annatto, carotenoids, caramels, and turmin were used to provide the desired color in sausages but meat alternatives generally also require a change in color after cooking from red to brown. Therefore, the use of natural heat-stable coloring agents is not appropriate. However, according to Plant-Ex Ingredients Ltd. (an international company in United Kingdom specialised in the manufacture of ingredients for color, flavor, and extracts), the red beet is an ideal coloring agent for products like burgers and minced meat products (as incorporated in a burger from The Beyond Burger company), as it gives a strong pink–red color to the raw product and after heating, the sugar in the beet caramelises and turns a shade of brown. In Figure 2A, an uncooked alternative sample with red beet colorant is shown, and Figure 2B shows the sample after cooking which displays the change in color from red to brown [16]. This change in color explains the objective of combining the reducing sugar (maltose, dextrose, lactose, mannose, or arabinose) with the heat-labile colorant [17]. The other method is the use of a myoglobin alternative. Food companies such as Impossible Foods are using a recombinantly produced soy leghaemoglobin from fungus and beetroot extract to mimic the color of raw meat [15,16]. Before the cooking process, the leghaemoglobin provides a fresh meat color to the product; however, after cooking, a color change takes place because of the Maillard reaction.

Another method has also been used which involves the use of colorants with protein-containing materials after injecting the mixture into an extruder barrel, i.e., the structuring process. Still, the final color obtained by all these processes is not of the highest quality, owing to the fact that different intrinsic and extrinsic factors affect the color of food, like the presence of maltodextrin and alginate hydrate, and the PH. The PH of meat alternatives can differ due to the addition of various acids like acetic acid and citric acid while formulation methods also affect the final color profile [17].

Recently, advanced technologies like artificial intelligence and machine learning have been revealing potential approaches in this area to mimic the color of meat products by conducting thorough investigation of the molecular structure and analysing the ways in which each component behaves independently and in the presence of each other [18]. The company Eat Just has developed and even marketed a plant-based egg alternative made from an Indian legume [19]. The use of other technologies like bioinformatics analysis with proteomics and sarcoplasmic protein is also receiving attention as these can be used in the formation of a better-colored product [20].

## 3. Texture Profiling of Animal and Plant-Based Meat

The textural profile (firmness, juiciness, springiness, and cohesiveness) is the main parameter for the quality and acceptability of a product and consumers are even willing to pay more to receive a product with better texture [21]. For a product to be considered meat, the correct fiber structure of the product is necessary. In order to develop an alternative product with a similar structure, prior analysis of the structure of meat is important. The size of a typical muscle fiber is between 1 and 40 mm in length and 20 and 100 μm in diameter; however, up to now the structural fiber of alternatives is produced at the micron level only [22,23].

The textural properties of muscle meat mainly depend upon the composition and structure of muscle fibers that are made up of myofibrils (actin and myosin filaments) along with connective tissue, sarcomere length, intramuscular fat, and denaturation of protein during cooking [24,25,26]. Other than these factors, non-meat ingredients like fats/oils, binders, dietary fibers and additives also affect the tenderness of meat. The textural and rheological properties of PBMAs can also vary from one product to another according to the ingredients used and method of processing just as meat from different parts of an animal has different attributes. So, according to the properties desired in the end product, an appropriate selection of raw ingredients and technique is necessary to mimic the fibral, connective, and adipose structure of the imitated product. The texturization of an alternative is basically the process (usually the extrusion) of rearranging the protein ingredients to form a fibrous structure that can mimic the technical and functional aspects of a meat. As it is well known that vegetable protein is a globular-shaped complex multimer while meat is fibrous in nature, the difference in the structure of these two elements is the biggest challenge to producing an alternative. The development of a fibrous structure during processing involves the occurrence of different events of unfolding, crossing, breakdown, and gelatinisation. The plant protein undergoes various processing techniques like extrusion, spinning, and shear force to be texturized by unfolding and cross-linking [27,28]. The protein not only provides nutrition but also contributes to other essential functional properties of the product through emulsifying, gelling, and water/oil-absorbing capacity.

Choosing the right protein source is important for the development of PBMAs. The most commonly used protein sources for PBMAs are soy protein concentrates and isolates as they are cost-effective and easily available, and have a similar mouthfeel and texture to meat after hydration [21]. The purified forms of protein obtained by fractionation of soy flour contain about 70 and 90% protein, respectively. It has also been observed that soy protein extenders can improve the water-holding capacity, chewiness, and juiciness of beef [22]. Palanisamy et al., in their study, found an improvement in textural profile caused by increasing the iota-carrageenan concentration in soy protein. Combining soy protein with wheat gluten (insoluble protein) also enhances the formation of layered and fibrous structures in PBMAs [23]; since the use of a single plant protein produces a weak structure, combining different proteins can effectively improve structure formation. Wheat gluten helps in binding and acts as a stabilizing agent while providing nutritional, swelling, binding, and structural properties [24,25]. Researchers have also observed that incorporating 30% gluten in the preparation results in the highest degree of texturization, chewiness, hardness, and fibrous texture. Various legumes such as peas, lentils, and beans are also used in the fabrication of PBMAs. Pea protein can be used as a substitute for soy protein which can cause allergies in some people and it is also cost-effective and easily available; however, structurization using pea protein is challenging as it has low gel-forming availability and also fewer useful functional and sensory properties [26,27]. Another protein source is oilseeds as they are rich in protein and can be utilised as a functional ingredient. Rapeseed mainly contains two proteins, cruciferin and napin, which initiate gel formation at high temperatures and pressure which aids in the texturization of PBMAs [28,29]. Quinoa flour has also been used as a gelling agent and fat replacer as it improves the nutritional properties and reduces cooking loss [30]. Other than plant-based meat alternatives, insects are also currently being studied by researchers as a suitable alternative to meat. Starowicz et al. have provided detailed information about edible insects as a meat alternative [31].

Apart from the plant protein, there are some constructional ingredients that help in improving the texture profile of PBMAs. As we have already mentioned, the incorporation of wheat gluten and iota carrageenan improves the fibrous structure [23,32]. Additionally, other research found that 3% methylcellulose with texturized soy protein has a similar texture, including cohesiveness and springiness, to beef patties [33]. Another water-soluble dietary fiber, Konjac glucomannan (β−1,4-linked D-mannose and D-glucose), acts as an emulsifier and stabilizer in alternative meat production [34]. An enzyme derived from the streptoverticillium moberansae, Transglutaminase, is also used as a cross-linking agent in alternative production [35,36].

Two main types of techniques are used in the processing of alternative products: top-down and bottom-up structuring techniques. In the top-down techniques (extrusion, freeze structuring, shear cell, and the blending of hydrocolloids), a fibrous anisotropic structure is obtained by using external force on biopolymer blends [37,38,39,40]. In bottom-up techniques, individual fibers are combined to form an end product [41] through wet spinning and electrospinning. The ingredients used in textured vegetable protein obtained from different sources like cereals, legumes, and oilseeds not only have different nutritional properties but also have different functional properties like gelling, forming, emulsifying, and water- and oil-holding capacity [42]. Their transformation during extrusion depends upon different parameters such as those related to operation like the feed composition and geometrical factors of the machine (die dimension and type, screw design), and the process variables (moisture, temperature, and speed of the screw).

## 4. Flavor Profile of Animal and Plant-Based Meat

The flavor of meat equates to its taste and eating experience. It is the combination of stimulation caused by different bioactive compounds in the taste receptors of human oral and nasal cavities [38]. Components like carbohydrates, fat, and protein act like the precursors of meat flavor. There are more than 1000 flavor components present in meat which are responsible for the specific taste of the meat. Raw meat does not have any aroma and only has the flavor of blood, metal, and salt; however, when it undergoes a heating process, due to complex decomposition, oxidation, reduction, and different chemical reactions, various volatile compounds (alcohol, alkenes, aldehydes, ketones, ester, acids, and ether) are produced to generate the meat-like flavor [39,40,41]. Three main thermal reactions occur in meat products during cooking that develop this specific meaty flavor. (1) The Maillard reaction, where the flavor precursors are reducing sugars, free amino acids, and peptides which lead to the development of pyrazine, and heterocyclic and sulfhydryl compounds (4 mercapto-5-methyl-3(4H)-furanone, 2-methyl-3-furanthiol, 2-methyl-3-methylthiofuran, 2-methyl-3-methyldithiofuran). (2) The degradation reaction which utilizes thiamine to produce thiols, sulfides, and disulfides. (3) The oxidation reaction which uses lipids and fatty acids to produce aldehydes, furans, unsaturated ketones, and aliphatic hydrocarbons [43]. Because of these reactions, meat has three major flavor profiles, i.e., a meaty flavor which is derived from the amino acids and water-soluble reducing sugar in the meat, a species-specific flavor due to differences in the composition of fatty acids and aromatic water-soluble compounds, and off-flavors which develop because of the oxidation of lipids and other degradation process during storage and processing.

Meanwhile, PBMAs which mainly contain soy and legume protein exhibit flavors that are a bit astringent and bitter because of the presence of compounds like phenols, saponin, isoflavones, and phenolic acids and they also have a beany taste due to the lipid oxidation and protein denaturation caused by lipid oxidation while processing [42,43]. Therefore, to achieve a meat-like flavor and mask the natural flavor profile of plant protein, the removal of unwanted flavors during processing (extrusion and shear processing of protein) by the extraction and deactivation of lipoxygenase is necessary [44] alongside the addition of various flavoring agents (natural and synthetic). These flavoring agents are displayed in Table 1.

The high-pressure and moisture-extrusion processes also alter the flavor of the meat because of changes in the constituents of flavor and dissemination of water, and also the alternation in the conformation of the protein. Additionally, the use of synthetic flavoring agents is also problematic as it reduces the quality of the product and can contain harmful components. To impart a meat-like flavor, various plant-based ingredients like natural spices, yeast extract, and hydrolyzed vegetable protein (HVP) are also widely used in the formulation of meat alternatives. HVP is a nutritional food additive which is prepared from a variety of plant proteins like soybean, corn, wheat, etc. and is broken down into small peptides and amino acids using acid hydrolysis (which can form carcinogenic compounds) or enzymatic hydrolysis (the mainly used method) under mild PH and temperature conditions [53,54,55]. Due to the presence of many volatile components such as pyridines, pyrrole, organic acids, furans, furanones, sulfur-containing compounds, alcohols, ester and phenols, treatment with sulfur compounds, reducing sugar, and yeast autolysis, they produce a strong meat-like flavor [56]. However, the optimization of flavor is still a big challenge in the production of alternatives that taste like meat.

## 5. Sensory Evaluation Methods Used in Analysis

Sensory evaluation is a set of scientific techniques used to measure, analyse, and interpret the human response to the properties and components of food as recognised by taste, smell, touch, appearance, and hearing [57]. Different tests are used in different conditions to understand the sensory profiles of food. Three types of sensory analysis are mainly used in the evaluation of plant-based products: subjective analysis, descriptive analysis, and the discrimination test. The most commonly used method is the nine-point hedonic scale also known as the consumer acceptability test or subjective analysis; it provides information about the overall acceptance and rejection of a product based on definite sensory properties like appearance, taste, texture, and flavor by an untrained participant or consumers [58]. This test is based on the likes and dislikes of consumers and gives the producer an understanding of the desirability of the product. The consumer preference data also help to increase the overall profile quality of the product by informing the variation of different formulation parameters during processing. However, analysis by the descriptive sensory method (objective analysis) gives a more elaborate assessment of the sensory profile of a product based on a predefined scale. Here, the qualitative and quantitative intensity of the sensory properties is determined.

Meat alternatives are rated according to different attributes like elasticity, fibrousness, brittleness, tenderness, juiciness, taste, flavor, appearance, and smell [24,59,60,61,62]. To accomplish this, panellists should be extensively trained on aspects including texture profile, flavor profile, quantitative descriptive analysis, and sensory spectrum [58]. The information obtained from this analysis quantifies these factors and contributes to the development of a better product. There is also one more sensory analysis test which is the discrimination test, used to detect the difference in the sensory parameters of two or more products. This test is performed in various ways. In the triangle test, three different unknown samples are given to the panellist and they have to identify the dissimilar sample among them. Meanwhile, in the duo–trio test, the panellist is given three samples; one is known and two are unknown. They have to match the given sample to the known one based on its sensory properties. However, in the ABX test, three samples are given to the panellist, two known and one unknown, and they have to match the unknown sample with one of the known samples. Table 2 is given to provide the sensory evaluation results of different plant-based alternative products obtained in different studies. This table provides an overview of the methods used in the sensory evaluation of plant-based products by different researchers.

## 6. Consumer Preferences and Acceptance

Our food habits not only affect our health but also the future of our planet. The production of meat in recent years has played a major role in climate change. According to Thomas et al. [73], the global anthropogenic greenhouse gas emissions from agricultural activity are about 30% of the total, with 14.5% coming from livestock. Over the past couple of years, consumers have also changed their behaviors toward eating healthy products, driving the need to develop alternative sustainable products. Eating habits pertaining to meat alternatives also depend upon social standards and education level; for instance, people tend to change their eating preferences to match their peers [74]. The increasing variety of plant-based alternatives on the market is providing a choice to the end consumer to replace the meat in their diet. According to these trends, plant-based meat alternatives would be more acceptable to non-meat consumers [75] than meat eaters. In addition to eating preferences, gender also plays a role in the acceptance of alternative products. In one study [76], maleness is positively correlated with the consumption of mammal muscle meat; however, females are more accepting of vegetarian and vegan diets than males. There are several factors that positively affect the consumption of meat alternatives but the primarily one is environmental concerns. Still, there are some sensory obstacles like dissimilar taste and texture in regards to meat, cost, color, and convenience limiting the acceptance of alternative products [77]. Non-meat eaters and meat eaters both think that meat alternatives should have more nutritious components like protein and vitamins, and less calories together with affordable pricing.

## 7. Challenges and Future Research Prospectives

Currently, plant-based proteins do not fully satisfy consumers in terms of quality, which is necessary for a product to qualify as an alternative. One basic criterion for a plant-based product to become an alternative to meat is to have a meat-like texture and water-binding ability. As plants and animal muscles have differences in protein type, amino acid composition, chemical composition, and sequencing of peptides, it is difficult to reproduce the same product. Beef’s most significant structural property is its capacity to bind with water (which contributes to the juiciness) and the structure of its muscle fiber; however, in alternative meat, the low moisture after cooking typically meets with disapproval from consumers [78,79,80,81]. A study was done to investigate the effect of buddle size on tenderness in bovine muscle. It was found that bovine muscles with smaller buddle size exhibit more tenderness after initial compression and less chewing force than the samples with larger size. Intramuscular fat (IMF) also positively affects the tenderness [82]. In the case of PBMAs, taste and textural profile can be enhanced by marination and impregnation [83,84,85,86]. The marinade’s composition could also affect the WHC of the final product [87,88]. This water-holding capacity (WHC) of a product can be described by a cross-linked polymer network theory known as Flory–Rehner theory [89,90]. This theory relates the WHC of the polymer network to material properties like polymer–water affinity and density. The Flory–Rehner theory can also describe the WHC of simplified meat analogues and how the marinade’s PH and ionic strength affect it, as this is not completely understood by researchers [91]. Different types of thickeners, water-binding agents, and texture enhancers are used in the processing of alternative products to obtain the properties of beef. Other parameters can be modified to mimic the taste and color of animal meat [92]. Soy proteins have a bitter, astringent flavor due to the naturally occurring saponin and isoflavone [93]. Furthermore, off-flavor can also be formed by the effect of heat on sugar and amino acids through the thermal degradation of thiamine. However, the bitterness of peas is mainly related to the saponin content, which depends on the variety of pea [94]. Hexanal is responsible for a hay-like off-flavor in frozen peas. Additionally, sulphur-containing compounds and aliphatic and aromatic hydrocarbons also contribute to the off-flavor of pea. To incorporate the flavor of processed meat and to mask their own taste, a whole range of herbs and spices are added at the time of processing. These flavoring agents include sodium chloride, potassium chloride, soy sauce, cane sugar, molasses, lactose, mannitol, vinegar, onion powder, celery, yeast extract, garlic, liquid smoke, black pepper, sage, oregano, paprika, rosemary, walnut, lemon juice, and others [95]. In addition to masking the off-flavor of the raw ingredients, there are certain methods that can also be used to remove the off-flavor before processing such as soaking and thermal treatment, germination, solvent extract, and fermentation. These methods have been extensively described [96]. Recently, gelatin and alginate-based hydrogels have been reported as thermoresponsive carriers for flavor [97]. Emulsion-filled calcium alginate gel beads (EF-CAGs) can be prepared by combining an O/W emulsion and alginate solution and then injecting them into a calcium ion solution. These hydrogels are basically complex structures in which lipid droplets are trapped in cross-linked biomolecules [98]. Different studies have confirmed that these are effective carriers for encapsulating bioactive agents to control their release in a simulated gastrointestinal environment by modifying their structures and properties [99,100]. However, recently, several researchers have evaluated the effect of heat treatment on the physical properties of calcium alginate beads; they found that the core materials can be released by physical changes [101,102,103,104], establishing that encapsulation of emulsion-filled microgels is useful to control the release of allyl methyl disulfide, a major flavor in garlic, during simulated cooking. The construction of nano emulsions and semi-solid gels has been successful to some extent for the flavor enhancement of alternative products but it must be further investigated to study the ingredient functionality.

The basic requirement for a product that is similar to meat is to have the same structural building blocks as meat which include myofibrils, muscle cells, and muscle tissue. To develop this structure from plant-based protein, and other components like the connective tissue layer to surround these myofibrils, binding agents can be used. The study of ingredient functionality is necessary to obtain a meat-like taste and develop a technique for processing plant-based fiber. Future investigations should focus on the fiber characterization present in alternatives with respect to the size, geometry, and interaction between them but this aspect has not yet been sufficiently developed. In addition, more advanced technology and tools should be developed that can analyze texture more quickly and without destroying the product. Further investigation should be performed to understand the changes in the textural properties of food after chewing or digestion.

## 8. Conclusions

Our existing protein source system has severely affected human health, the planet, and animals. Plant-based alternative products are a healthier and more sustainable way to counter all these issues but at the moment, plant-based alternatives do not match all the parameters of animal products like nutritional level, sensory aspect, and cost. Improvement of these parameters to produce the best possible alternative will help to maintain the market size of alternative products. As we know, based on the ingredient composition and processing method, plant-based alternatives have different nutritional and sensory properties. It is essential to carefully study the sensory properties of a product in reference to the texture, color, and taste, as these are the parameters that are mainly hindering the consumption and consumer acceptability of meat alternatives. The field of plant-based meat alternative development requires the cooperation of all the related companies, scientists, and economic experts to successfully tackle these issues.

## Figures and Tables

**Figure 1 foods-13-00108-f001:**
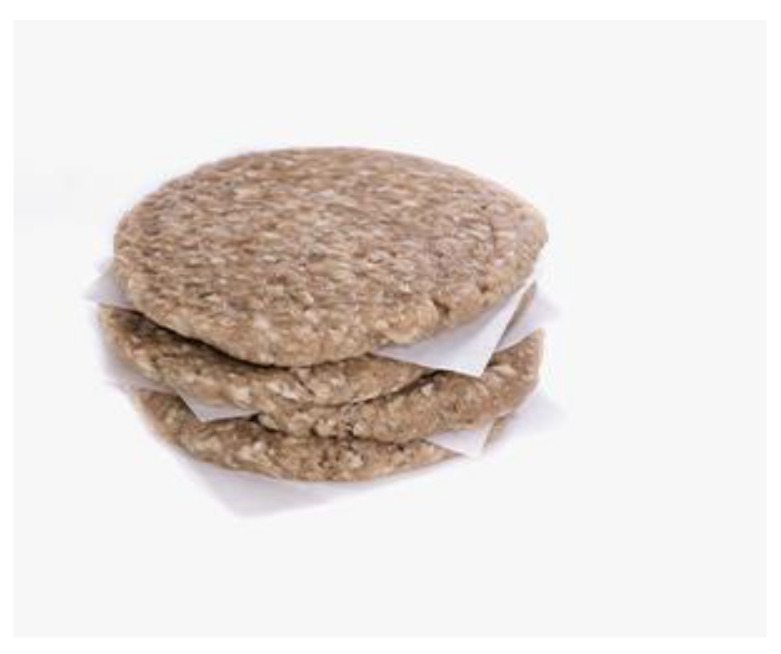
Representation of natural plant-based alternative color profile.

**Figure 2 foods-13-00108-f002:**
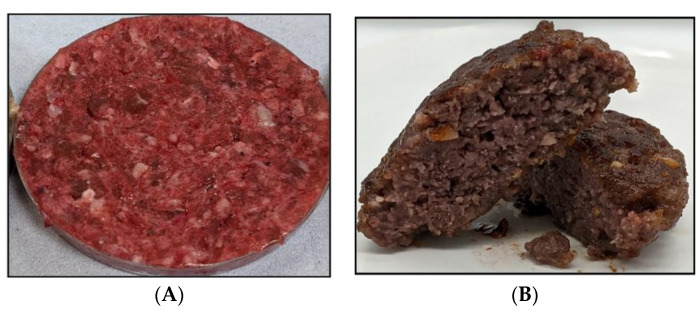
Illustration of change in color (Red Beet) of plant-based patty. (**A**) Red Beet Pre-Cooking. (**B**) Red Beet Post-Cooking.

**Table 1 foods-13-00108-t001:** Types of different natural and synthetic flavoring agents.

Name	Natural Flavoring Agents	Synthetic Flavoring Agents
Definition	Natural flavoring agents are substances that are extracted from plants, herbs, spices, and microorganisms. [45]	Synthetic flavoring agents are substances that are similar to natural agents. [45]
Type	Herbs and spices: Garlic, onion [46,47]	Artificial: Artificial smoke flavor or synthetic versions of natural flavors [48]
	Yeast Extract [49,50]	Flavor enhancer: Monosodium glutamate (MSG)
	Fermented products: Miso, tamari [51]	
	Vegetable Extract: Tomatoes, mushrooms [45,52]	
Cost	Expensive [45]	Less expensive [45]

**Table 2 foods-13-00108-t002:** Examples of different studies on the sensory evaluation of different alternative products. The abbreviations used in the table are as follows: SPI (Soy Protein Isolate), PBA (Plant-Based Alternative).

Product Type	Method	Panelist	Finding	Reference
PBA to chicken Nuggets	Consumer acceptability	105 untrained	PBA does not have fibrous structure and has beany or off-flavor	[63]
Peanut-based alternative to beef patty	Consumer acceptability	60 untrained	The sensory properties were better than the soy-based alternative and it can be a consumer-accepted substitute for beef patty	[64]
Chicken sausage (SPI)	Descriptive analysis	8 trained	The sausage was equally acceptable in terms of overall acceptability	[65]
PBA to beef patty	Descriptive analysis	10 trained	No beany essence was noticed	[4]
Sausage analogue (mushroom-based and SPI)	Consumer acceptability	32 untrained	Mostly closed characteristics like beef (can be applied as a substitute)	[66]
Meat Analogue (defatted soy, rice and bean flour)	Odor and color (after sous-vide treatment)	73 untrained	Color scores were higher in analogue than beef	[67]
Meat analogue (Oat–pea protein)	Hedonic (appearance, taste, odor)	8 trained	Highly fibrous structure with mild flavor	[68]
Chicken analogue (SPI and Wheat gluten	Consumer acceptability	unknown	Highly fibrous structure in comparison to chicken breast	[24]
Meat analogue (faba bean protein)	Sensory and instrumental analysis	unknown	The product had good bite-feeling, elasticity/firmness in comparison to meat	[69]
PBA to chicken and beef	Consumer acceptability	71 untrained	Meaty flavor and juiciness are absorbed	[70]
PBA (Gluten-free and soy-free)	Consumer acceptability	60 untrained	The addition of anthocyanins increases the antioxidant capacity of the product with an acceptable color change	[71]
Meat analogue	Consumer acceptability	93 untrained	Similarity to meat does not seem to have an effect on the acceptance	[72]

## Data Availability

Data is contained within the article.

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
