# Peer review of "Sensory Evaluation of Plant-Based Meat: Bridging the Gap with Animal Meat, Challenges and Future Prospects"

_foods, 2023, doi:10.3390/foods13010108_

Round 1
Reviewer 1 Report
Comments and Suggestions for Authors
This study explores the comparative differences in the sensory profile of animal meat and plant-based meat alternative specially in the color, texture and flavor, along with the method used to resemble them. Additionally, a direction to further research for developing better plant-based alternative products has also been suggested. While the analysis of this paper is general and lacks specific analysis content, the authors should more specifically highlight the novelty of the study. The language also needs to be improved. Please refer to the comments in the attached file.

The language also needs to be improved.
Author Response
Reviewer:1
This study explores the comparative differences in the sensory profile of animal meat and plant-based meat alternative specially in the color, texture and flavor, along with the method used to resemble them. Additionally, a direction to further research for developing better plant-based alternative products has also been suggested. While the analysis of this paper is general and lacks specific analysis content, the authors should more specifically highlight the novelty of the study. The language also needs to be improved. Please refer to the comments in the attached file.
Comments:
- Line 68: Formatting Error
Response:
There is a formatting error in the Line 68. Therefore, it has been corrected in the revised manuscript.
“The”
- Line 75: This part seems to only compare some red meat, is there any research in other meat products or processed meat? Lack of literature summary and classification.
Response:
The comparison is based on the alternative product that have been marketed. As most of the plant based alternative produce are in the patty and sausage form and at the moment not any other form of product is marketed for the consumers. We have added some more information on about the colouring agents.
“ For decades the synthetic food colouring agents have also been widely used in PBMA. However, customers find them undesirable due to side effects include toxicity, allergy, and neurocognitive consequences. Hence the naturally occurring pigment like anthocyanin, carotenoids, curcuminoids, used as colouring pigment.”
- Line 102: the statement is not clear, please rephrase it.
Response:
As the reviewer suggested, the authors have rewritten the statement and also added additional information for the better understanding.
“However, According to Plant-Ex Ingredients ltd. (an international company specialised in the manufacturing food ingredients from colours, flavours and extract) the red beet is an ideal colouring agent for the products like burgers and minced meat products (as incorporated in burger from The Beyond Burger company) as it gives a strong pink-red colour to the raw product and after heating the sugar in the beet caramelise and gives a shade of brown.”
- Line 149: what is PBA?
Response:
PBA is the plant-based alternative for the convenience we have changed it to “PBMA” (plant-based meat alternative) throughout the manuscript for the better uniformity.
- Line 163: what specific plant protein can be used? The content of this part is not detailed enough and the research progress of the structure and function of the plant proteins has not been shown. Besides plant protein, are there other ways to improve the texture of plant protein, a lot of research is now focusing on insect protein.
Response:
As suggested by reviewer we have added more information regarding the use of different plant protein.
“Choosing a right protein source is important for the development of PBMA. The most commonly used protein source for the PBMA is soy protein concentrate and isolate as they are cost effective, easily available and have the similar mouthfeel and texture like meat after hydration[21]. The purified form of protein by fractionation of soy flour contains about 70 and 90% protein respectively. It has also been observed that the soy protein extenders can also improve the water holding capacity, chewiness and juiciness of beef[22].Palanisamy et al. in their study found the improvement in textural profile by increasing the iota-carrageenan concentration in soy protein. Combining soy protein with wheat gluten (insoluble protein) also enhance the formation of layered and fibrous structure in PBMA[23] as the use of single plant protein produce a weak structure, combining different protein can effectively improve the structure formation. The wheat gluten helps in the binding and act as stabilising agent while providing nutritional, swelling, binding and structural property[24, 25]. Researchers have also observed that incorporating 30% of gluten in the preparation have the highest degree of texturization, chewiness, hardness and fibrous texture. Various legumes such as pea, lentil, beans are also used in the fabrication of PBMA. The pea protein can be a substitute for the soy protein as it can be allergic to some people and it is also cost effect, easily available however the structurization using pea protein is challenging it has low gel forming availability and also less feasible functional and sensory property[26, 27]. Another protein source is oilseeds as they are rich in protein and can be utilised as a functional ingredient. The Rapeseed mainly contains two proteins, cruciferin and napin which initiate the gel formation at high temperature and pressure which aids to the texturization of PBMA[28, 29]. Quinoa flour is also been used as gelling agents and fat replacer as it improves the nutritional property and reduce the cooking loss[30].
For the letter part of the question about the insect protein. Yes, now the insect protein is only used in the preparation of meat alternative however as this review is solely focusing about the development of plant-based meat alternative we don’t think its is necessary to include the information. Considering reviewer comment on this we have added a statement about it and also the reference of a paper for detailed information.
“Other than plant-based meat currently insects are also been studied by researcher as a suitable alternative of meat. Starowicz et al. have provided the detailed information about edible insect as alternative[31].”
Apart from the plant protein there are some constructional ingredients that helps in improving the texture profile of PBMA. As we have already mention that incorporation of wheat gluten and iota carrageenan improve the fibrous structure[23, 32]. Additionally, in research by, they found that the 3% methylcellulose with texturized soy protein have similar texture including cohesiveness and springiness as of beef patties[33]. An another water-soluble dietary fiber Konjac glucomannan ( β−1,4-linked D-mannose and D-glucose) acts aa an emulsifier and stabiliser in alternative production[34]. A n enzyme derived from the streptoverticillium moberansae, Transglutaminase is also used as a crosslinking agent in alternative production[35, 36]”.
- Line 185: what are the typical flavor compounds in meat products? For example, what aldehydes, ketones, ester substance, they may come from lipid hydrolysis oxidation, milliard reaction, protein hydrolysis, oxidation etc.. familiarity with the flavor substances of typical meat products can be learned form the meat flavors.
Response:
The typical flavour compounds of meat include flavour compound such as 4 mercapto-5-methyl-3(4H)-furanone, 2 -methyl-3-furanthiol, 2-methyl-3-methylthiofuran, 2-methyl-3-methyldithiofuran, unsaturated ketones., hexane, pentane thiols, sulfides and disulfides. And the main reaction that take place in these flavour developments are thermal, oxidation and degradation reaction. The information has been added into the revised manuscript.
“Three main thermal reactions occur in meat products during cooking that developing this specific meaty flavour. 1). The Maillard reaction where the flavour precursors are reducing sugars, free amino acids, peptides which leads to the development of pyrazine, heterocyclic and sulfhydryl compounds (4 mercapto-5-methyl-3(4H)-furanone, 2 -methyl-3-furanthiol, 2-methyl-3-methylthiofuran, 2-methyl-3-methyldithiofuran). 2). The degradation reaction which utilizes the thiamine to produce thiols, sulfides and disulfides. 3) the oxidation reaction by using lipids and fatty acids to produce aldehydes, furans, unsaturated ketones, aliphatic hydrocarbons[43]”.
- Line 194 how does lipid oxidation occurs in proteins?
Response:
We have corrected the sentence for better understanding.
“Due to the lipid oxidation and protein denaturation caused by lipid oxidation “
Here we were referring to the denaturation of protein that is stimulated by lipid oxidation. As the oxidation of lipid formed free radicals and other molecules like ketones and aldehydes. And these free radicals attack the main chain and side chain of protein causes the protein oxidation and deterioration.
- Line 199: What are the specific applications? These should be relatively mature in the commercialization; it is suggested that the author can use the chart to illustrate.
Response:
These agents are used to improve the flavour profiling of the PBMA like in sausage and patties made for plant-based protein ingredients. As suggested by reviewer the authors have added a table explaining these flavouring agents.
|
Name |
Natural flavouring agents |
Synthetic flavouring agents |
|
Definition |
Natural Flavouring agents are the substance that are extracted from plant, herbs, spices and microorganism. [45] |
Synthetic flavouring agents are substances that are similar to natural agents [45] |
|
Type |
Herbs and spices: Garlic, onion [46, 47] |
Artificial: artificial smoke flavour or synthetic version of natural flavours[48] |
|
Yeast Extract [49, 50] |
Flavour enhancer: Monosodium glutamate (MSG) |
|
|
Fermented products: Miso, tamari [51] |
||
|
Vegetable Extract: Tomatoes, mushrooms [45, 52] |
||
|
Cost |
Expensive [45] |
Less expensive [45] |
- Line 203: What good results do these applications get?
Response:
“HVP is a nutritional food additive which is prepared from a variety of plant protein like soybean, corn, wheat etc broken down into small peptides and amino acids using acid hydrolysis (can form carcinogenic compounds), enzymatic hydrolysis (mainly used method) under mild PH and temperature[45-47]. Due to the presence of many volatile components such as pyridines, pyrrole, organic acids, furans, furanones, sulfur containing compounds alcohols, ester, phenols treatment with sulfur compounds, reducing sugar and yeast autolysis they produce a strong meat like flavour[48].”
Currently the manufactures use HVP to intensify the flavour of food products like soups, gravy, sauces, stews, stocks and processed meat products. This is also used to restore the loss of flavour during canning, freezing or drying process.
- Table: No finding
Response:
The authors have added the finding for this part of the table.
- Line 236: There seems to be a lack of analytical literature on these sensory evaluation s in table 1.
Response:
In the table 1 the authors have tried to summaries the consumer acceptability of the PBMA for the products available for the consumption and this table is simply explain the finding about the prospective of consumer acceptance. the authors have also added some more information. For the discrimination test of analysis specific data is not available for the PBMA.
|
Chicken analogue (SPI & Wheat gluten |
Consumer acceptability |
Unknown |
High fibrous structure in comparison to chicken breast |
[24] |
|
Meat analogue (faba bean protein) |
Sensory and instrumental analysis |
Unknown |
The product had good bite-feeling, elasticity/firmness in comparison to meat |
[65] |
- Line 262: Can the authors tell us what the specific reasons other than environmental factors.
Response:
The production of PBMA is positively affected by certain factors like resource efficiency (as the production of PBMA require fewer natural resources like water, land compared to the traditional meat), animal welfare (ethical concerns about the animal treatment), health consideration (as the alternative offers lower levels of saturated fats and cholesterol) but mainly by the environmental concerns.
- Line 271: one basic criterion
Response:
This has been corrected as the reviewer suggested “One basic criterion”.
- Line 299: How to ensure the nutritional content of plant-based meat products such as essential amino acids, lysine and how to control the cost should be considered.
Response:
Certain factors can be considered to ensure the nutritional profiling of PBMA:
- Selection of ingredient: the ingredient used in the production should be rich in essential amino acids. Other than that, combining different plant proteins like legume, pulse seeds can also help in the balancing amino acid profile. For example: lentils or chickpeas with quinoa.
- Fortification of product: this involves the addition of isolated amino acid into formulation of the product or simply using fortified raw ingredient.
For the costing point of view: To reduce the cost of the final PBMA.
- the sourcing of ingredient that are available locally and exploring the bulk purchase option.
- Optimization of the production process to minimize the byproduct waste.
- Increasing the production volume can also reduce the cost per unit.
- Continuously investing in research to find innovative, cost-efficient ingredients and process.
As the goal of this review is to study the sensory profile of the PBMA we have not added about the nutritional profiling and costing of the product.
- Line 327 The format of the reference is confused.
Response:
The authors have modified the reference throughout the manuscript and also updated the references list.
- Minor editing of English language required
Response:
As mentioned by reviewer we have made change to improve the English in the manuscript.
“According to a report by Bloomberg in 2021 [1], globally the plant-based food market is expected to reach up to $162 billion by 2030 from $29.4 billion in 2020 because as the concept of flexitarian (a causal vegetarian) is getting quite popular in today’s generation with 14% vegetarians or vegan and 15% flexitarians, which is approximately 29% of total global population [2]. Additionally, the people those are suffering from different health related issues like lactose intolerance or malabsorption, high cholesterol intake are also finding their ways to plant based alternative products [3]. “

Reviewer 2 Report
Comments and Suggestions for Authors
Line 39: Currentely, --------- till Line 45: this paragraph require reference.
Line 62: you mentioned (taste) without percentage> Also, please explain the diff. % as it is not 100% in their sum>
Line 77: Delete 9upon to) and replace by (depends on)>
Line 103: plant-Ex ingredients ltd (Declare this).
Line 227: in table 1: Abbreviations should be declared>
Aso, all references in table 1 are not present in References section>
Comments on the Quality of English LanguagePlease improve the quality of english editing in your paper.
Author Response
The authors are thankful to the reviewers for their consideration of our manuscript and we appreciate their very valuable suggestions, which have been very helpful in refining the manuscript. All comments received have been taken into account to improve the quality of our article and we present a point-by-point response to each comment.
Reviewer 2:
- Line 39: Currentely, --------- till Line 45: this paragraph require reference.
Response:
As suggested by the reviewer the authors have added the reference of the paragraph and also specified the location for which the data is stated.
- Line 62: you mentioned (taste) without percentage> Also, please explain the diff. % as it is not 100% in their sum>
Response:
About 82% of the consumer believe that the taste is the most important factor that effect the purchase power of a product. The data is taken form a survey where the consumers were asked denote the parameter which influence their purchasing power. These data are suggesting that the taste is most important factor after that the cost, health, convenience and lastly how the product production affect the environment sustainability.
- Line 77: Delete 9upon to) and replace by (depends on)>
Response:
The change has been made as per the suggestion by the reviewer.
“depends on “
- Line 103: plant-Ex ingredients ltd (Declare this).
Response:
The Plant-Ex Ingredients ltd. Is an international company specialised in the manufacturing food ingredients from colours, flavours and extract. the authors have rephrased the sentence with additional information.
“However, According to Plant-Ex Ingredients ltd. (an international company specialised in the manufacturing food ingredients from colours, flavours and extract) the red beet is an ideal colouring agent for the products like burgers and minced meat products (as incorporated in burger from The Beyond Burger company) as it gives a strong pink-red color to the raw product and after heating the sugar in the beet caramelise and gives a shade of brown,”
- Line 227: in table 1: Abbreviations should be declared> Aso, all references in table 1 are not present in References section.
Response:
The authors have declared the abbreviation mentioned in the table and also updated the reference of table in the references section.

Reviewer 3 Report
Comments and Suggestions for Authors
The paper is only informative and very simply written without sufficient scientific contribution and explanation. The most crucial aspect - nutritional aspect, i.e. the chemical composition section is missing. There are many terminological errors, especially for sensory evaluation.
Comments on the Quality of English LanguageModerate editing of English language required
Author Response
The authors are thankful to the reviewers for their consideration of our manuscript and we appreciate their very valuable suggestions, which have been very helpful in refining the manuscript. All comments received have been taken into account to improve the quality of our article and we present a point-by-point response to each comment.
Reviewer 3
- The paper is only informative and very simply written without sufficient scientific contribution and explanation. The most crucial aspect - nutritional aspect, i.e. the chemical composition section is missing. There are many terminological errors, especially for sensory evaluation.
Response:
The authors are agreed with the reviewer point that the nutritional profile of PBMA is important however as the review only focus on the sensory profile especially texture, colour and flavour we have not included the nutritional aspect of the PBMA. Moreover, to improve the informative point of view of manuscript we have added more information throughout in different sections.
“Choosing a right protein source is important for the development of PBMA. The most commonly used protein source for the PBMA is soy protein concentrate and isolate as they are cost effective, easily available and have the similar mouthfeel and texture like meat after hydration[21]. The purified form of protein by fractionation of soy flour contains about 70 and 90% protein respectively. It has also been observed that the soy protein extenders can also improve the water holding capacity, chewiness and juiciness of beef[22].Palanisamy et al. in their study found the improvement in textural profile by increasing the iota-carrageenan concentration in soy protein. Combining soy protein with wheat gluten (insoluble protein) also enhance the formation of layered and fibrous structure in PBMA[23] as the use of single plant protein produce a weak structure, combining different protein can effectively improve the structure formation. The wheat gluten helps in the binding and act as stabilising agent while providing nutritional, swelling, binding and structural property[24, 25]. Researchers have also observed that incorporating 30% of gluten in the preparation have the highest degree of texturization, chewiness, hardness and fibrous texture. Various legumes such as pea, lentil, beans are also used in the fabrication of PBMA. The pea protein can be a substitute for the soy protein as it can be allergic to some people and it is also cost effect, easily available however the structurization using pea protein is challenging it has low gel forming availability and also less feasible functional and sensory property[26, 27]. Another protein source is oilseeds as they are rich in protein and can be utilised as a functional ingredient. The Rapeseed mainly contains two proteins, cruciferin and napin which initiate the gel formation at high temperature and pressure which aids to the texturization of PBMA[28, 29]. Quinoa flour is also been used as gelling agents and fat replacer as it improves the nutritional property and reduce the cooking loss[30]. Other than plant-based meat currently insects are also been studied by researcher as a suitable alternative of meat. Starowicz et al. have provided the detailed information about edible insect as alternative[31].
Apart from the plant protein there are some constructional ingredients that helps in improving the texture profile of PBMA. As we have already mention that incorporation of wheat gluten and iota carrageenan improve the fibrous structure[23, 32]. Additionally, in research by, they found that the 3% methylcellulose with texturized soy protein have similar texture including cohesiveness and springiness as of beef patties[33]. An another water-soluble dietary fiber Konjac glucomannan ( β−1,4-linked D-mannose and D-glucose) acts aa an emulsifier and stabiliser in alternative production[34]. A n enzyme derived from the streptoverticillium moberansae, Transglutaminase is also used as a crosslinking agent in alternative production[35, 36]”.
For the terminological errors authors have changes some terminologies in the table. “Descriptive analysis”
- Moderate editing of English language required
Response:
As suggested by reviewer we have rephrase sentence for better understand in the revised manuscript
“However, it is exceedingly difficult to replicate the texture and flavour of meat due to the differences in the molecular and physiochemical properties of plants and animals as well as the lower protein efficiency and lack of nutrients in plant materials.”
“This review aims to provide a better understanding of the sensory differences between plant-based and animal-based meat alternatives by comparing their colour, texture, and flavour profiles. It also discusses the techniques used in their development and the sensory evaluation to develop the consumer preferred alternative in both the qualitative and quantitative way.”

Reviewer 4 Report
Comments and Suggestions for Authors
Comments to the Authors
The review entitled " Sensory Evaluation of Plant-based Meat: Bridging the Gap with Animal Meat, Challenges and Future Prospects", submitted by Kumari et al. is focused on the discussion of color, texture, and flavor profile of plant-based meat alternative in comparation to animal meat. Researchers’ intention is to provide a better understanding about the sensorial aspects around plant-based meat alternative and meat products. The topic of this review is novel and interesting. Unfortunately, the topic that the authors intend to publish has already been addressed recently in some of the excellent published reviews in the Foods journal. For example, a complete review about textural properties of plant-based alternative (PBA), including meat alternatives, has been addressed by Moss et al. (2023) (Moss, et al. 2023. A Prospective Review of the Sensory Properties of Plant-Based Dairy and Meat Alternatives with a Focus on Texture. Foods, 12(8), 1709). Also, these authors included an extensive review of sensory evaluation methods used in PBA evaluation. Fiorentini et al. (2020) (Fiorentini, et al. (2020). Role of sensory evaluation in consumer acceptance of plant-based meat analogs and meat extenders: A scoping review. Foods, 9(9), 1334), included an extensive review related to the color and overall appearance, taste, flavor, aroma, and texture of plant-based meat alternatives. Also, these authors included a review about the sensory evaluation methods used in the evaluations of PBA. Int relation to the challenges and future prospects, Alcorta et al. (2021) (Alcorta, A.; Porta, A.; Tárrega, A.; Alvarez, M.D.; Vaquero, M.P. Foods for Plant-Based Diets: Challenges and Innovations. Foods 2021, 10, 293), included a complete revision of this topic. Other reviews that addressed important aspect included in the manuscript submitted by Kumari et al, published only in Foods journal in the last two year are the following:
Yu, J.; Wang, L.; Zhang, Z. Plant-Based Meat Proteins: Processing, Nutrition Composition, and Future Prospects. Foods 2023, 12, 4180. https://doi.org/10.3390/foods12224180
Sengar, A.S.; Beyrer, M.; McDonagh, C.; Tiwari, U.; Pathania, S. Effect of Process Variables and Ingredients on Controlled Protein Network Creation in High-Moisture Plant-Based Meat Alternatives. Foods 2023, 12, 3830. https://doi.org/10.3390/foods12203830
Safdar, B.; Zhou, H.; Li, H.; Cao, J.; Zhang, T.; Ying, Z.; Liu, X. Prospects for Plant-Based Meat: Current Standing, Consumer Perceptions, and Shifting Trends. Foods 2022, 11, 3770. https://doi.org/10.3390/foods11233770
Zahari, I.; Östbring, K.; Purhagen, J.K.; Rayner, M. Plant-Based Meat Analogues from Alternative Protein: A Systematic Literature Review. Foods 2022, 11, 2870. https://doi.org/10.3390/foods11182870
Szenderák, J.; Fróna, D.; Rákos, M. Consumer Acceptance of Plant-Based Meat Substitutes: A Narrative Review. Foods 2022, 11, 1274. https://doi.org/10.3390/foods11091274
Kołodziejczak, K.; Onopiuk, A.; Szpicer, A.; Poltorak, A. Meat Analogues in the Perspective of Recent Scientific Research: A Review. Foods 2022, 11, 105. https://doi.org/10.3390/foods11010105
In my opinion, the review submitted by Kumari et al. has already been extensively addressed by other authors and published in the last two years.
Regardless of the reasons that this reviewer has given, if the editor-in-chief considers that the topic proposed by the authors is of interest to the scientific community, then the manuscript should be substantially improved. Several points of clarification and suggestions are indicated below.
General: All sections of the manuscript need to deepen the information
Line 12. Please, address only the topic related to plant-based meat.
Line 22. Please, address only the topic related to plant-based meat.
Line 28. The introduction section needs to include more information about sensory aspects of PBA and meat products, especially those about color, texture, and flavor profile. These are the topics of the review.
Line 208. Delete the symbol (:) at the end of the title
Lines 209-246. I think it makes no sense to address sensory evaluation methods, since they are the traditional ones for evaluating any food. Probably, of greater interest would have been to compile results from different studies highlighting the effects of certain ingredients or processing conditions on sensory parameters such as color, texture, flavor, etc.
Line 226. Table 1 needs to be improved. Please see some examples in published paper in Foods journal. Also, the reference in Table 1 should be according the author`s guide.
Line 255. Correct the coma after the word “instance”
Line 271. Word “One”, in uppercase
Lines 277-279. Include some studies addressing this topic.
Lines 282-283. Include some studies addressing this topic.
Lines 283-285. Include some studies addressing this topic.
Line 327. References should be according to the author`s guide.
Author Response
The authors are thankful to the reviewers for their consideration of our manuscript and we appreciate their very valuable suggestions, which have been very helpful in refining the manuscript. All comments received have been taken into account to improve the quality of our article and we present a point-by-point response to each comment.
Reviewer 4:
- Line 12.Please, address only the topic related to plant-based meat.
Response:
As suggested by the reviewer the authors have change the plant-based alternative to” plant-based meat”.
- Line 22.Please, address only the topic related to plant-based meat.
Response:
As suggested by the reviewer the authors have change the plant-based alternative to” plant-based meat”.
- Line 28. The introduction section needs to include more information about sensory aspects of PBA and meat products, especially those about color, texture, and flavor profile. These are the topics of the review.
Response:
The authors understood the point made by reviewer however in the introduction part the authors want to point out the need to improve the sensory profile of the PBMA and its effect on the consumer acceptance. While considering reviewer’s comment the authors have added additional information the colour, texture and flavour section particularly denoted to that profile.
- Line 208. Delete the symbol (:) at the end of the title.
Response:
The symbol (:) have been deleted throughout the manuscript.
- Lines 209-246. I think it makes no sense to address sensory evaluation methods, since they are the traditional ones for evaluating any food. Probably, of greater interest would have been to compile results from different studies highlighting the effects of certain ingredients or processing conditions on sensory parameters such as color, texture, flavor, etc.
Response:
In this manuscript the authors have tried to sum up the researches that have produces different type of alternative product for the meat as these products are mainly the processed product likes patty, sausage. By this table the authors want to summaries the consumer acceptability of the PBMA for the products available for the consumption and this table is simply explain the finding about the prospective of consumer acceptance. To highlight the effect of ingredient of the sensory property we have added more information.
“The most commonly used protein source for the PBMA is soy protein concentrate and isolate as they are cost effective, easily available and have the similar mouthfeel and texture like meat after hydration[21]. The purified form of protein by fractionation of soy flour contains about 70 and 90% protein respectively. It has also been observed that the soy protein extenders can also improve the water holding capacity, chewiness and juiciness of beef[22].Palanisamy et al. in their study found the improvement in textural profile by increasing the iota-carrageenan concentration in soy protein. Combining soy protein with wheat gluten (insoluble protein) also enhance the formation of layered and fibrous structure in PBMA[23] as the use of single plant protein produce a weak structure, combining different protein can effectively improve the structure formation. The wheat gluten helps in the binding and act as stabilising agent while providing nutritional, swelling, binding and structural property[24, 25]. Researchers have also observed that incorporating 30% of gluten in the preparation have the highest degree of texturization, chewiness, hardness and fibrous texture. Various legumes such as pea, lentil, beans are also used in the fabrication of PBMA. The pea protein can be a substitute for the soy protein as it can be allergic to some people and it is also cost effect, easily available however the structurization using pea protein is challenging it has low gel forming availability and also less feasible functional and sensory property[26, 27]. Another protein source is oilseeds as they are rich in protein and can be utilised as a functional ingredient. The Rapeseed mainly contains two proteins, cruciferin and napin which initiate the gel formation at high temperature and pressure which aids to the texturization of PBMA[28, 29]. Quinoa flour is also been used as gelling agents and fat replacer as it improves the nutritional property and reduce the cooking loss[30]. Other than plant-based meat currently insects are also been studied by researcher as a suitable alternative of meat. Starowicz et al. have provided the detailed information about edible insect as alternative[31].
- Line 226. Table 1 needs to be improved. Please see some examples in published paper in Foods journal. Also, the reference in Table 1 should be according the author`s guide.
Response:
As suggested by reviewer the authors have added additional sensory evaluation from the paper published by Foods journal.
|
Chicken analogue (SPI & Wheat gluten |
Consumer acceptability |
Unknown |
High fibrous structure in comparison to chicken breast |
[24] |
|
Meat analogue (faba bean protein) |
Sensory and instrumental analysis |
Unknown |
The product had good bite-feeling, elasticity/firmness in comparison to meat |
[65] |
- Line 255. Correct the coma after the word “instance”
Response:
The coma after the word instance has been corrected.
- Line 271. Word “One”, in uppercase
Response:
This has been corrected as “One basic criterion”.
- Lines 277-279. Include some studies addressing this topic.
Response:
As suggested by the reviewer the authors have added additional information.
“A study was done to find out the effect of buddle size on the tenderness in bovine muscle. Where they found that the bovine muscles with the smaller buddle size exhibits more tenderness after initial compression and less chewing force than the bovine with larger size. And also the Intramuscular fat(IMF) also positively affect the tenderness[73]. While in the case of PBMA the taste and textural profile can be enhanced by the marination and impregnation[74-77]. And The marinade composition could also affect the WHC of final product[78, 79]. This water holding capacity (WHC )of a product can be describe by a cross-linked polymer network theory known as Flory-Rehner theory[80, 81]. This theory relates the WHC of the polymer network to material properties like polymer- water affinity and density. The Flory- Rehner theory can also describe the WHC of simplified meat analogue and how the marinade PH and ionic strength effect it as it is not completely understood by the researchers[82]”.
- Lines 282-283. Include some studies addressing this topic.
Response:
As suggested by the reviewer the authors have added additional information.
“Furthermore, off- flavour can also be formed by the effect of heat on sugar and amino acids, by thermal degradation of thiamine. However, the bitterness of peas is mainly related to the saponin content, which depend on the variety of pea[85]. Hexanal is responsible of hay-like off-flvour in the frozen peas. Additionally, the sulphur containing compounds and aliphatic and aromatic hydrocarbons are also contributed in the off-flavouring of pea.”
- Lines 283-285. Include some studies addressing this topic.
Response
As suggested by the reviewer the authors have added additional information.
“These flavouring agents include sodium chloride, potassium chloride, soy sauce, cane sugar, molasses, lactose, mannitol, vinegar, onion powder, celery, yeast extract, garlic , liquid smoke, black pepper, sage, oregano, paprika, rosemary, walnut, lemon juices and others[86]. Other than masking the off-flavour of the raw ingredients there are certain methods that can also be used to remove the off-flavour before processing such as soaking and thermal treatment, germination, solvent extract, fermentation. These methods have been extensively described[87]. Recently, gelatin- and alginate-based hydrogels have been reported as thermoresponsive carriers for flavour[88]. The emulsion-filled calcium alginate gel beads (EF-CAGs) can be prepared by combining an O/W emulsion and alginate solution and then injecting them into a calcium ion solution. These hydrogels are basically complex structure in which lipid droplets are trapped in crosslinked biomolecules[89]. Different studies have confirmed these are effective carriers for encapsulating bioactive agents to control their release in a simulated gastrointestinal environment by modifying their structures and properties [90, 91]. However recently, several researchers have evaluated the effect of heat treatment on the physical properties of calcium alginate beads; they found that the core materials can be released by physical changes [92-95]establish that encapsulation of emulsion-filled microgels is useful to control the release of allyl methyl disulfide, a major flavour in garlic, during simulated cooking.”
- Line 327. References should be according to the author`s guide.
Response:
The authors have modified the reference throughout the manuscript and also updated the references list.

Round 2
Reviewer 1 Report
Comments and Suggestions for Authors
accept
Reviewer 3 Report
Comments and Suggestions for Authors
Accept in present form
Comments on the Quality of English LanguageMinor editing of English language required.